# Walking around the Autonomous Province of Trento (Italy): An Ethnobotanical Investigation

**DOI:** 10.3390/plants11172246

**Published:** 2022-08-29

**Authors:** Valeria Cavalloro, Francesco Saverio Robustelli della Cuna, Elena Quai, Stefania Preda, Francesco Bracco, Emanuela Martino, Simona Collina

**Affiliations:** 1Department of Earth and Environmental Sciences, University of Pavia, Via Sant ’Epifanio 14, 27100 Pavia, Italy; 2Department of Drug Sciences, University of Pavia, Via Taramelli 12, 27100 Pavia, Italy

**Keywords:** ethnobotany, Trentino—Alto Adige, Italy, RFC

## Abstract

The Trentino-South Tyrol region is a special statute region of northeastern Italy. This territory is of particular interest for its morphology, flourishing vegetation, and history, having been a meeting area among different civilizations. Hence, Trentino is characterized by an ethnic plurality and a rich ethnobotanical knowledge, even if the available information is fragmentary, widely dispersed, and often guarded in oral popular culture. To fill this gap, in the present work 200 subjects were interviewed using an ethnobotanical survey. The resulting 817 citations referred to 64 native species, used either for human or animal health or for domestic purposes. As a second step, for each plant exploited for medicinal purposes, local importance was evaluated by calculating their relative frequency of citation. Moreover, the main traditional preparations were discussed. Among them, the most cited and exploited ones are *Achillea millefolium*, *Arnica montana*, *Hypericum perforatum*, *Malva sylvestris*, *Pinus mugo,* and *Satureja montana*, for which a deeper analysis has been performed. Lastly, the ethnobotanical knowledge of the plants growing in this territory will add a piece to the mosaic of traditional medicine in Italy and may lay the foundation for a nature-aided drug discovery process.

## 1. Introduction

Trentino-South Tyrol region is a special statute region of northeastern Italy bordering Austria and Switzerland. It is a mountainous region where different Alps sections meet (Dolomites, Alpi Retiche, Alpi dei Tauri, and Prealpi), reaching 3905 m a.s.l. with its highest peak named Ortles. Trentino-South Tyrol, is also characterized by several valleys and it is crossed by rivers (the most important one is the Adige river) and lakes (e.g., Garda lake in the south). Due to this peculiar and varied territory, Trentino-South Tyrol is characterized by very rich biodiversity. Thus, near Garda Lake, many Mediterranean species grow, while hornbeam, beech, and maple trees prevail in the north up to an altitude of 1200–1400 m. Furthermore, spruce, larch, and birch trees prevail in higher altitudes, but give way to Alpine pastures and tundra vegetation above 2000 m. Finally, valleys are the perfect environment to harvest species like apple trees. Besides this high richness in terms of biodiversity, Trentino-South Tyrol is also a crossroads of different populations and ethnicities. Thus, being located in the northernmost part of Italy, Trentino-South Tyrol has always been characterized by an ethnic plurality, having been a meeting area among different civilizations, strongly influencing the local culture. Taken together, both these peculiar characteristics make Trentino-South Tyrol territory a very interesting hotspot of traditional knowledge, mainly belonging to the ethnobotanical field [1,2]. During the centuries, ethnobotanical knowledge has been mainly handed down orally both vertically, among generations, and horizontally, among members of the same community. In many cases, the absence of written findings and the progressive departure from traditions are causing the loss of traditional knowledge of natural remedies. This statement can be considered true for all the Italian ethnobotanical traditions and, for this reason, always more scientific papers have as final aim to rekindle attention on this topic. Consistently, starting from 1981, many ethnobotanical investigations have been carried out in different Italian areas (Table 1), with the common purpose of collecting information to achieve greater awareness of the history of a certain location. As highlighted in Table 1, none of the ethnobotanical investigations published until now has the Trentino-South Tyrol region as subject.

The present work aims to shed light on the ethnobotanical knowledge of Trentino Alto Adige. So far, the available information is fragmentary, widely dispersed, and often guarded in oral popular culture or Italian written [32,33,34]. Particularly, we selected and interviewed 200 subjects of different sex, age, and occupation to collect and analyze ethnobotanical information in the medicinal, veterinary and liquoristic fields. The knowledge of traditional uses of the plants growing in this territory will add a piece to the mosaic of traditional medicine in Italy. Moreover, the most relevant collected data may represent the starting point for a nature-aided drug discovery process, one of the most winning approaches to identify new active ingredients able to contrast human diseases [35,36,37,38].

## 2. Results and Discussion

### 2.1. Studied Area and Data Collection

Trentino-South Tyrol is a region with special status in northeastern Italy. It borders Austria to the north, Veneto to the east, Lombardy, and Switzerland to the west. The territory is mountainous and is characterized by numerous valleys. It hosts about 1,050,000 inhabitants with a density of 73 inhabitants per km² and has an area of 13,607 km². Trentino-South Tyrol is composed of two autonomous provinces, Trento and Bolzano. The Province of Trento or Trentino is divided into 11 sectors, corresponding to the old districts: Val di Sole, Valle di Non, Valli Giudicarie, Alto Garda and Ledro, Valle dell’Adige, Vallagarina, Alta Valsugana, Bassa Valsugana and Tesino, Valle di Fiemme, Primiero and Ladino di Fassa (Figure 1). These districts differ both in their morphology and the level of tourism.

Considering that people are the repository of knowledge and information, we paid particular attention to the selection of the informants, an essential step for performing a correct ethnobotanical investigation. Informants have been selected based on knowledge of traditional medicinal plants, and age. Overall, from June 2019 to December 2020, two hundred people have been interviewed, out of these 133 (67%) were female and 67 (32%) were male. The informants were categorized into five different age groups, as outlined in Table 2. The informants were aged from 18 to 87 years, and the average age was 53 years. In contrast with other areas [39], the majority of the informants were female, probably as a consequence of Trentino industrialization, which prompted most of the males to leave the agricultural sector, thus leaving back ethnobotanical customs. Conversely, many females have kept the job of housewives, looking after the home and children, and teaching the ethnobotanical traditions.

The recruitment of informants involved all the 11 areas of Trentino. As outlined in Table 3, data collection has been more difficult in heavily tourist areas, such as Val di Fassa, Alta Valsugana, or the Primiero, where the knowledge of the traditional uses of medicinal plants was lower, maybe because of the intense work in tourist places, which brought the inhabitants to move away from their traditions. Thus, in these areas, a maximum of 63 citations (Val di Fassa) have been collected from 12 informants. Different considerations can be made for a few tourist areas, as can be highlighted by the data related to Valle di Non and Valle di Fiemme, where a similar number of informants, 19 and 15, is associated with a very higher number of citations, 175 and 148, respectively. Informants have been interviewed following the common principle in ethnobotany, i.e., what, where, when, who, why, and how. All data have been collected, reporting the name of the plants, their medical use, parts used, methods for preparation, and collection time. Overall, 817 citations were obtained, out of these 65 were referred to native species of the Autonomous Province of Trento.

### 2.2. Plant Species Cited by Informants

Firstly, we performed a literature search on the 817 cited plants and classified them on their use: plants for human health (87.1%, 712), plants for preparation of liqueurs (11.5%, 94), and plants for veterinary use (1.4%, 11) (Figure 2). The cited officinal natural sources include one lichen (*Cetraria islandica* L.) which is now an emerging organism in the drug development field [40] and plants belonging to 31 different families, among which the most represented ones are Asteraceae (10 plants), Pinaceae (5 plants), and Lamiaceae and Rosaceae (4 plants each) families.

We particularly focused on the 56 species specifically mentioned by informants for medicinal uses and calculated relative frequency citation (0  <  RFC  <  1), useful to identify the frequency of use and the local importance of each species. The RFC index is given by the frequency of citation (FC, the number of informants mentioning the use of the species) divided by the total number of informants participating in the survey (N), without considering the use categories [41]. A full list of cited plants (scientific name, habitat, plant part used, etc.) is reported in Appendix A.

As shown in Table 4, *Achillea millefolium* L. (RFC = 0.1), *Arnica montana* L. (RFC = 0.47), *Hypericum perforatum* L. (RFC = 0.35), *Malva sylvestris* L. (RFC = 0.17), *Pinus mugo* Turra (RFC = 0.34), *Satureja montana* L. (RFC = 0.135), and *Urtica dioica* L. (RFC = 0.11) are the most exploited medicinal plants, as the high RCFs values highlighted. Conversely, the less frequently exploited biomasses were *Cetraria islandica*, *Cornus mas*, *Cyclamen sp.pl.*, *Elymus repens*, *Juniperus sp.pl.*, *Lycopodium clavatum*, *Senna alexandrina*, * Solanum nigrum*, and *Verbascum thapsus* with an RCF value of 0.005. All these plants were cited only once a time by the informants.

Traditionally prepared formulations of cited plants are employed by local people. The most common traditional preparations mentioned by the informants are herbal tea, decoction, tincture, syrup, oleolyte, and ointment. They are briefly discussed in the following paragraph.

### 2.3. Traditional Preparations

Informants reported different therapeutic applications of plants, depending on their route of administration. Consistently, the main preparation methods for oral administration (herbal tea and decoction) or topical administration (oleolyte and ointment) are described here below, whereas the plants most used in this preparation are reported in Table 5.

Herbal tea is an aqueous preparation extemporaneously prepared to administer orally plants for therapeutic purposes or used as a vehicle for other medicaments [42]. Briefly, the dried vegetal part is added to boiling water and left to infuse for ten minutes, keeping the container covered. At the end of the infusion, the solid is separated using a strainer of a paper filter. Another extemporaneous preparation is a decoction, commonly used to extract active compounds from barks, branches, leathery leaves, roots, and rhizomes. Extraction of active ingredients from such plant materials requires prolonged boiling times. Informants described two different decoction methods. Most of the informants put drugs and water in the container at the same time, then bring everything to a boil for a few minutes. Few informants (especially elderly people) add the drug to boiling water and let the mixture boil for several minutes. In both cases, the solid must be separated from the water by filtration, thus obtaining the decoction. The tincture is a liquid preparation obtained by maceration of fresh plant material in ethyl alcohol at 90° for at least 40 days. Some informants tend to leave the drug as soon as it is collected for a day in the shade in a warm place before the tincture preparation so that the biomass is as dry as possible. Oleolyte is a topical formulation prepared by maceration of the fresh plant materials in vegetable oil (olive, sunflower, or rice oil). Such a formulation combines the emollient properties of the oils with the therapeutic activity of the ingredients extracted from the medicinal plants [42]. The fresh plant material is placed into a jar with a lid together with the oil and left to macerate for forty days, mixing from time to time, and then the plant materials are separated from the oil by filtration. Finally, the oil enriched with active metabolites so obtained can be exploited both topically and orally (the traditional in-house packaging is pictured in Figure 3 panel A,B). From the interviews emerged that great attention must be paid to the humidity of the plant material used for preparing oleolytes. Some informants avoid collecting plant material in the morning but wait until the early afternoon, to prevent the presence of dew and therefore to limit the presence of water, others do not use the fresh plant but prefer to leave them under the sun (or in the shade in a warm place) for a day before maceration to reduce humidity. Some others, during the maceration, leave the cap of the jar slightly unscrewed and every two or three days dry the condensation that forms on the surface of the jar with paper. All the above-mentioned measures are aimed to reduce the presence of water, which could lead to microorganisms’ growth and the contamination of finished products. An example of oleolyte commonly used by local people for alleviating gout and joint pain is obtained by *Rhododendron ferrugineum*’s galls (or cecidia). Oleolytes may be used as such or maybe an ingredient of an ancient topical preparation called ointment.

The vernacular name of ointment is “ünt del tai”, which translation is “the grease of the cut”. Since ancient times the woodcutters collected the oleoresin (a honey-like substance) of Larix decidua, called “Largà” in the Trentino dialect, and applied it to the hands’ wounds produced during their work (Figure 3 panel C). Over the years, Largà has been used to extract wooden thorns or treat wounds with abscesses and local men exerted the ancient trade of “Largaiòl”, i.e., turpentine collector. The harvest involves carving the Larix trunk, creating a hole about 10 cm from the ground which is immediately closed with a cap. After three years (time limit by law [43]), the cap is removed and the Largà is collected in buckets.

Once harvested, Largà is traditionally prepared using:-1/3 oleoresin (Largà);-1/3 raw beeswax;-1/3 oleolyte of choice;-Essential oil or alcohol 90°

Briefly, the raw beeswax is melted till the wax is liquefied, and then Largà and the oleolyte were added. The so obtained mixture was heated for a few minutes on the stove. Once the ingredients are well-blended, essential oil or alcohol at 90° (as a preservative agent) is added. The so-obtained preparation was conserved in glass jars. Informants use ointments to improve the performance of the oleolyte, combining the therapeutic properties of the medicinal plant with those of Largà. In cases of muscle sprains or contractures, the candidates recommend the use of *Arnica montana*-based oleolyte for anti-inflammatory activity but, since the application of the oleolyte is uncomfortable, they prefer to use the ointment that allows having both the therapeutic properties of the two species in the same preparation. The same concept can be adapted to *Hypericum perforatum*-based ointment in the case of burns, erythema, or dermatitis. Another example of ointment is a particular preparation called mugolio, which is a syrup obtained by resinous pine cones of *Pinus mugo*. It is traditionally prepared as Mugolio (Mugo + oil) starting from the buds collected in summer, following two different methods (Figure 3 panel D). The first one involves the stratification of pine cones or buds with sugar in a jar. The recipient is then exposed to the sun for at least 40 days, till the complete dissolution of the sugar, and then the mixture is filtered. The second method foresees the filling of the jar with pine cones or buds, the addition of honey, and the subsequent maceration under sunlight for at least 40 days. This preparation is used by local people against cough and sore throat.

### 2.4. A Survey of the Most Used Plants

As expected, plants with the highest RCFs values (Table 4) are the plants most used for the preparation of traditional remedies. In Table 5, the most relevant plants, their traditional preparation, and their medicinal uses are reported. The most cited and used medicinal plants are *Achillea millefolium*, *Arnica montana*, *Hypericum perforatum*, *Malva sylvestris* L., *Pinus mugo*, and *Satureja montana* (entries 1, 15, 19, 24, and 33).

Over the years, several reviews regarding these plants have been published, and they can be useful to readers interested in deepening the knowledge about the above-discussed medicinal plants [44,45,46,47,48,49]. Particularly, *A. millefolium* and *H. perforatum* have been used for both oral and topical remedies. Indeed, Informants highlighted that oral and topic preparations have completely different applications as shown in Table 5 (entries 1 and 15, respectively). These different applications are strictly related to the preparation methods. Thus, herbal tea and decoction allow the extraction of water-soluble secondary metabolites. Particularly, herbal tea is characterized by mild extraction conditions (the water is added after boiling and not during boiling, so the temperatures are low). As a consequence, if, on one hand, herbal tea preserves thermolabile metabolites, on the other one this preparation allows the extraction of few metabolites due to the short extraction time and low temperature. Conversely, decoction requires higher temperature and longer extraction times, compared to herbal tea. Topic preparation (ointment) foresees the extraction of metabolites by oils (apolar solvents) and therefore different secondary metabolites are extracted.

### 2.5. Most Cited Plants

*Achillea millefolium*, also called Yarrow, belongs to the Astraceae family. It is an herbaceous, perennial, and aromatic plant with a branched and creeping rhizome and straight stem. The top of the corymbs bears several white or pink flower blooms, while the species name millefolium is due to the leaves, which are very jagged in depth. *A. millefolium* grows in meadows, pastures, and along the edges of paths up to 2200 m a.s.l. The importance of this plant as a medicinal remedy can be found also by analyzing its generic name. Thus Achillea derived from the Greek hero Achille, who exploited this plant to treat his companions’ wounds. *A. millefolium* is traditionally used in the treatment of inflammation, gastrointestinal and gynecological disorders, hepato-biliary complaints, and wound healing [44]. Generally, the aerial parts are the ones exploited to prepare the tea, decoction, or oleolyte to treat different human disorders. Recent scientific studies confirmed the *A. millefolium* traditional uses. Thus, the choleretic effect is mainly due to dicaffeoylquinic acids, while the spasmolytic activity was attributed to the flavonoids [50]. In detail, apigenin and luteolin (both glycosylated and not) resulted to have very interesting antioxidant and anti-inflammatory activities, while the metabolites mainly present in the lipophilic preparations are β-pinene, sabinene, 1,8-cineole, and β-caryophyllene. The different hydro-lipophilic nature of the metabolites that are present in the phytocomplex extracted by *A. millefolium* may explain the different traditional preparations. Particularly, oleolyte (Table 5, #1) is able to extract lipophilic metabolites useful for treating muscle and joint pain (Figure 4), while decotion is exploited to extract apigenin, luteolin dicaffeoylquinic acids for treating urinary tract infections (Table 5, #1) [50,51,52]. No literature data supporting the use of herbal tea against menstrual pain and oligomenorrhea (Table 5, #1,) are available so far and therefore further investigations are needed.

*Arnica montana* is a perennial herbaceous plant of the Asteraceae family, formerly known as Compositae. It is characterized by radical rosette leaves, and an erect stem, which ends with a yellow inflorescence. In Trento Arnica is a well-known medicinal herb and grows spontaneously throughout the territory at different altitudes, even if the harvest is preferred in areas higher than 1500 m. From a biological point of view, the most interesting families of secondary metabolites produced by *A. montana* are sesquiterpene lactones of the pseudoguaianolide group (helenalin and its esters with short-chain fatty acids); caffeoylquinic acids (chlorogenic acid and derivatives); flavonoids (quercetin, kaempferol, patuletin, and their glycosides and hispidulin); essential oil (thymol and derivatives) (Figure 5) [53]. Thymol and cymene derivatives may explain the traditional use of ointment and oleolyte (Table 5, #3), being endowed with anti-hyperalgesic and anti-inflammatory activities [54].

The secondary metabolites profile is generally influenced by endogenous and exogenous factors; in this particular medicinal herb, the endogenous ones are the age of the plant, while the exogenous ones are the altitude and therefore the radiation, the geographical area, the soil, the attack of parasites and others [55]. In 2011 Huber and collaborators highlighted that the sesquiterpene lactones significantly decrease inflammation by inhibiting the NF-kB complex, a mechanism of action that differs from that of non-steroidal anti-inflammatory drugs, such as indomethacin and acetylsalicylic acid [56]. All the 94 citations referred to this plant are associated with pathologies involving the musculoskeletal system, such as contractures, sprains, or strains. Candidates benefit from the therapeutic properties of *Arnica* by extracting its active ingredients through various preparations, i.e., mother tincture called Arnica spirit, oleolyte or ointment, or Arnica flower pack. This latter case is exploited in urgency when fresh flowers are available. However, this method is rarely used because the extraction of the active ingredients is not optimal.

*Hypericum perforatum*, also called St. John’s wort, is a perennial herbaceous plant belonging to the Hypericaceae family. *H. perforatum* has a stiff and branchy stem, 30 to 50 cm high, and leaves arranged two by two against each other dotted with spots. The flowers, which bloom numerous at the apex of the branches, are bright yellow and have five petals. Flowering occurs from June to August depending on the altitude. This medicinal plant grows and develops in alpine pastures, it prefers uncultivated land, along the ditches or the edges of roads and woods up to 1600 m. The part of the plant with biological activity is represented by the flowering tops, rich in hypericin and pseudohypericin, biapigenin, hyperforin and derivatives, and cinnamic acids (Figure 6) [57]. Herbal tea (Table 5, #15), obtained by a common and traditional extraction process, is endowed with antidepressant activity. Based on literature evidence, this activity is due to a synergistic effect among hypericin hyperforin and its derivatives and some flavonols [58,59]. Moreover, most of the candidates exploit *H. perforatum* to treat skin pathology, preparing oleolites or ointments to be applied to injured skin (photosensitizing can be a side effect). This particular application is supported by several scientific papers, which highlighted that more lipophilic secondary metabolites like hypericin and hyperforin have anti-inflammatory and antimicrobial activities, as well as properties concerning the stimulation of growth and tissue differentiation [60]. Finally, in the Trentino tradition, *H. perforatum* is used to treat pathologies involving the respiratory system, such as inflammatory states of the bronchi. Moreover, in this case, an herbal tea prepared with its flowering tops is used.

*Malva sylvestris* belongs to Malvaceae family and it could be an annual, biennial or perennial herbaceous plant. It has an erect or prostrate stem that can grow from 60 to 80 cm. The palminervia leaves own 5 to 7 lobes and are characterized by irregular margins. The pink flowers are gathered at the axil of the leaves and bloom from April to October, they have dark streaks and bilobed petals. The fruit is a circular polychaene. As for *A. millefolium*, also the importance as a medicinal plant of *M. sylvestris* can be found in its name. Thus, “malva” means soft in Latin, highlighting its use as an emollient. *M. sylvestris* can be directly eaten to have a mild laxative, liver cleansing tonic, and anti-heartburn effects. On the other hand, its pharmaceutical preparations can be used to treat gastrointestinal and urological disorders, abdominal pain, diarrhea, respiratory diseases, insect bites, burns, furuncles, and ulcerous wounds [61]. The aerial parts of the plants are the most exploited, particularly the leaves and the flowers due to their anti-inflammatory properties. In detail, their traditional use fulfills the requirement for at least 30 years of medicinal use at a specified strength and specified posology. For this reason, the European Medicinal Agency and, more, in particular, the Committee on Herbal Medicinal Products (HMPC) assess that flowers of *M. sylvestris* can be used to treat mouth irritation or throat as well as mild stomach and gut pain in patients from 12 years of age [62]. Despite no specific metabolites have been recognized as responsible for *M. sylvestris* biological activities, the main class of secondary metabolites which are supposed to be active are flavonoids and mucilage. Candidates benefit from the therapeutic properties of *M. sylvestris* by extracting its active ingredients mainly by preparing herbal teas to treat dyspepsia (Table 5, #19).

*Pinus mugo*, or Mountain pine, is a needle-like bush of the Pinaceae family; this evergreen shrub, also known as dwarf pine, can reach up to three meters in height and is equipped with needles curved and crescent-shaped, rigid, from 3 to 8 cm long; the mountain pine has rather superficial but very branched roots and conical ovate cones, 2 to 5 cm long. This spontaneous shrub prefers mountainous areas, it is particularly widespread in the mountains of Trentino, where it grows in a range between 1500 m and 2600 m. In folk medicine, *P. mugo* has been exploited in various medicinal contexts, as an antitussive, and for lung diseases, as it boasts antiseptic, anti-inflammatory, expectorant, and fluidifying properties and secretolytic and antimicrobial effects [63,64]. *P. mugo* was cited 78 times by 200 informants, 69 citations reported for the treatment of diseases of the respiratory system, while 9 citations referred to liqueur use. Moreover, results highlighted that two preparations are used for the extraction of the active ingredients: Mugolio (see Section 2.4) and essential oil. The essential oils are produced by the distillation of the needles and their balsamic properties are used against the cold by applying a few drops on the chest (Table 5, #24) (Figure 5 panel D). The therapeutic properties of this traditional preparation are mainly attributed to monoterpenes like α-pinene, δ-3-carene, β-pinene, β-phellandrene (Figure 7) [48]. Finally, in the liqueur field, the pinecones of *P. mugo* are used for the production of grappa; the pine cones are left to infuse in the distillate for at least 40 days, in warm places under the sunlight, stirring occasionally.

*Satureja montana* is a suffruticose perennial plant with a strong aromatic odor of the Lamiaceae family. It is characterized by woody stems at the base, erect, usually widely branched from the bottom, to form a small bush with bright green, opposite and subsessile leaves. It is native to the mountainous regions of central-southern and western Europe that inhabits rocky arid soils up to 1300 m of altitude. This plant is exploited by informants to prepare herbal tea to contrast nervous gastric pains (Table 5, #33). The use of this specific traditional preparation finds an explanation in scientific publications. Thus, a paper published in 2010 assesses that the non-volatile extracts showed the strongest antioxidant activity compared to the essential oils obtained by the same biomass. This strong activity can be attributed to the high content of phenolic compounds. In the same work, the antimicrobial activity of *S. montana* essential oils and ethanolic extracts is investigated. In this second case, the activity resulted mainly due to phenolics (mainly carvacrol, and thymol) and terpenes (γ-terpinene) [65]. As mentioned for *A. montana*, also in the case of *S. montana* the active secondary metabolites concentration is strongly influenced by the environment. In detail, the concentration of its major components, e.g., carvacrol, linalool, terpinene, and cymene, shows large variations based on the harvesting place and the stage of development (Figure 8) [66]. Conversely, no explanations have been found in the literature for the activity against bloating, and vomiting of the herbal tea (Table 5, #33). Further investigations are needed.

## 3. Materials and Methods

### 3.1. Study Area

Trentino-South Tyrol region, a special statute region of northeastern Italy. It borders to the north with Austria, to the east with Veneto, and to the west with Lombardy and Switzerland. The territory is completely mountainous and is characterized by numerous valleys. It hosts about 1,050,000 inhabitants with a density of 73 inhabitants per km^2^ and an area of 13,607 km^2^.

### 3.2. Ethnobotanical Survey

From June 2019 to December 2020, the ethnobotanical use of plants was documented by interviewing 200 people, both men, and women of different ages. The study was explained to every informant before they gave oral informed consent. Each of them agreed to participate voluntarily and was allowed to discontinue the interviews at any time. The questionnaire, which was used as a tool for data collection, consisted of the following questions/authorizations:Informant data: Name and surname; place and date of birth; sex; education and work; source of the ethnobotanical knowledge (i.e., family, community, literature, or others).Authorizations: Authorization to exploit photos to certify the harvest and preparation of the plant; consensus to use the data provided by the informant.Plant information: Common (or scientific) and local name; collection place and time; traditional uses (medicinal, veterinarian, or liquoristic); part of the plant exploited and traditional preparation (infusion, food, decoction, …).Biological properties: Attributed property, target pathology, side effects (if any), other information.

Finally, the plants were recognized under the guidance of local people by studying available samples or by analyzing photos and comparing them to literature information.

### 3.3. Quantitative Data Analysis

The local significance of plants exploited for medicinal purposes was studied given the relative frequency of citation. The RCF values were calculated as follows:RCF = CFs/N
where CFs is the number of informants that have mentioned the use of the species and N is the total number of informants.

## 4. Conclusions

Trentino-South Tyrol is an Italian special statute region characterized by ethnic plurality, as testified by the fact that three languages are spoken there: Italian, German, and Ladin, a peculiar idiom whose origins are still unknown. During the decades, the different civilizations that met in this territory exploited the rich biodiversity of Trentino-South Tyrol in a different way, thus building a wide and well eradicated ethnobotanical culture. Unfortunately, all this knowledge has always been handed down orally and only a few fragmentary, incomplete, or Italian written works have been published. Consistently in the present contribution we fill this gap to rekindle attention on Trentino-South Tyrol ethnobotanical knowledge. In detail, 200 subjects were interviewed using an ethnobotanical survey. These informants were classified based on their district, bringing out that the most touristic areas have almost lost their ethnobotanical knowledge, due to the progressive removal of traditional habits. The main holders of this knowledge are women, probably as a consequence of Trentino industrialization, which prompted most males to leave ethnobotanical customs, while females continued to look after the home and children and teach the ethnobotanical traditions. The 817 citations resulting from the survey referred to 64 native species, used either for human or animal health or for domestic purposes. Moreover, based on the relative frequency citation index and the analysis of the most important traditional preparations, *Achillea millefolium*, *Arnica montana*, *Hypericum perforatum*, *Malva sylvestris*, *Pinus mugo*, and *Satureja montana* resulted to be the most exploited medicinal plants. The analysis of their traditional preparations highlighted that some of their applications were already known and the metabolites responsible for the activity identified. Conversely, some applications remain unjustified, like *A. millefolium* herbal tea to treat menstrual pain and oligomenorrhea or the same preparation of *S. montana* against bloating, and vomiting.

To sum up, Trentino-South Tyrol region, is characterized by ethnic plurality and it is a powerful source of ethnobotanical and ethnomedicinal knowledge. The available information belonging to the oral culture has been herein collected, thus avoiding the risk of being forgotten and lost. Moreover, the results highlight the promising role of medicinal plants in managing different human diseases. However, to confirm their therapeutic use, a deeper investigation is needed to approve the safety and efficacy of the extracts and their bioactive compounds.

## Figures and Tables

**Figure 1 plants-11-02246-f001:**
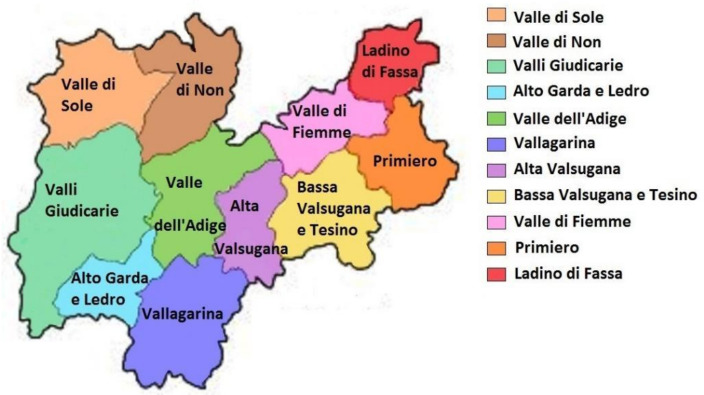
Map of the study area. Image adapted from Provincia Autonoma di Trento (https://www.provincia.tn.it/, accessed on 11 July 2022).

**Figure 2 plants-11-02246-f002:**
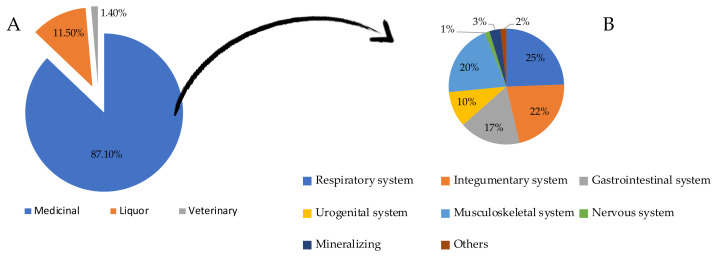
(**A**) Percentage of citations collected during the interviews grouped for categories of use, and (**B**) main applications of the medicinal plants.

**Figure 3 plants-11-02246-f003:**
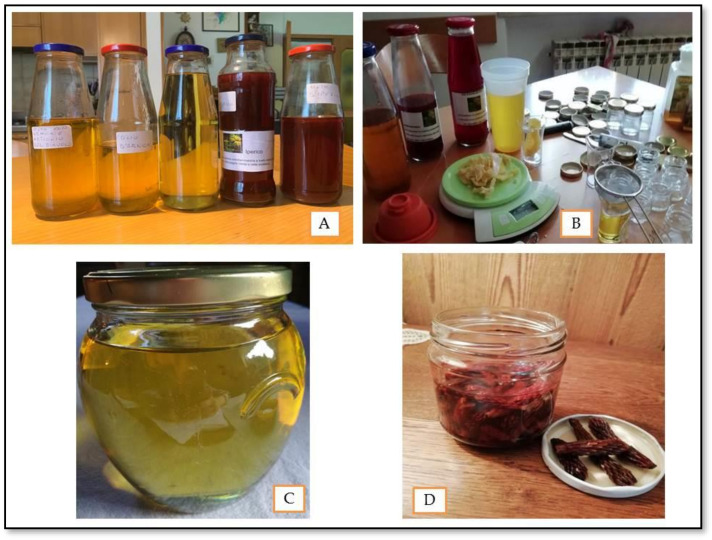
Common “in-house” packaging of traditional preparations: oleolyte (**A**,**B**), ointment traditionally called “ünt del tai” (**C**), and Mugolio (**D**).

**Figure 4 plants-11-02246-f004:**
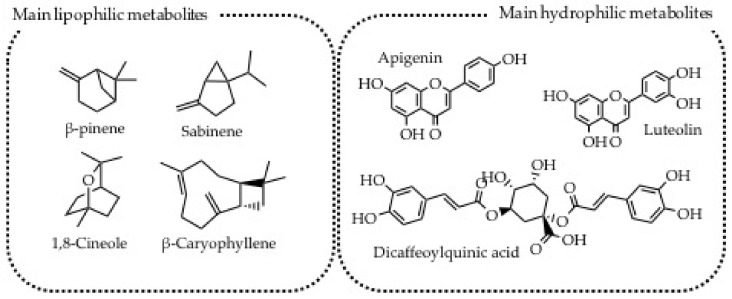
Most abundant secondary metabolites produced by *Achillea millefolium*.

**Figure 5 plants-11-02246-f005:**
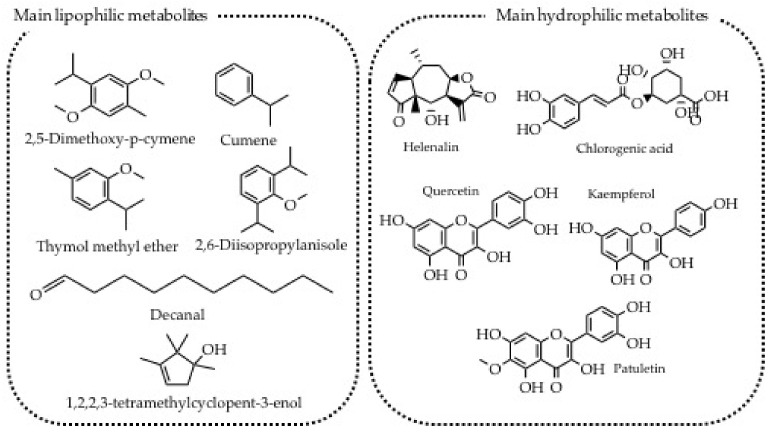
Main secondary metabolites produced by *Arnica montana*.

**Figure 6 plants-11-02246-f006:**
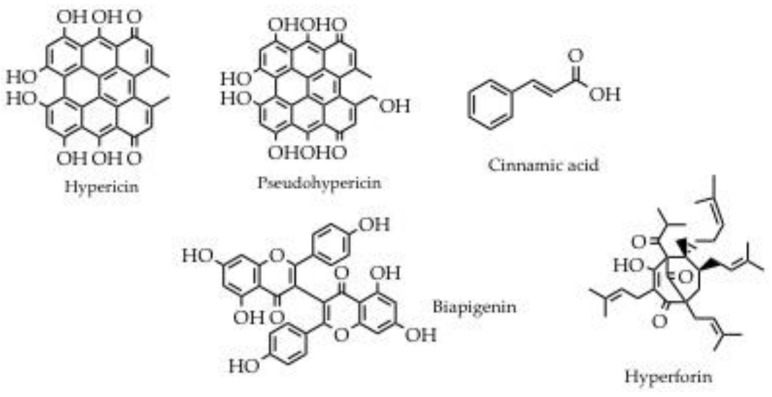
Main secondary metabolites produced by *Hypericum perforatum*.

**Figure 7 plants-11-02246-f007:**
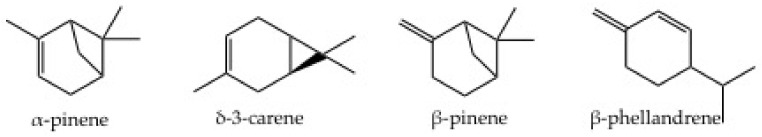
Main secondary metabolites produced by *Pinus mugo*.

**Figure 8 plants-11-02246-f008:**
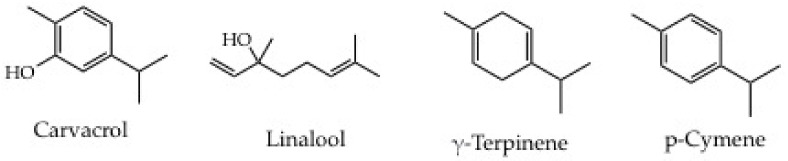
Main secondary metabolites produced by *Satureja montana*.

**Table 1 plants-11-02246-t001:** Ethnobotanical investigations carried out in different Italian areas from 1981 to 2022.

Author	Location	Plants (n°)	Informants (n°)
Ballero et al., 2001 [3]	Sardegna	65	105
Guarrera et al., 2005 [4]	Basilicata	56	49
Salerno et al., 2005 [5]	Basilicata	60	49
Guerrera et al., 2005 [6]	Lazio	96	44
Nebel et al., 2006 [7]	Calabria	48	54
De Natele et al., 2007 [8]	Campania	95	56
Guarrera et al., 2008 [9]	Molise	70	54
Signorini et al., 2009 [10]	Sardegna	72	17
Vitalini et al., 2009 [11]	Lombardia	58	54
Pieroni et al., 2009 [12]	Piemonte	88	67
Idolo et al., 2010 [13]	Lazio e Molise	145	60
Montesano et al., 2012 [14]	Basilicata	52	11
Leto et al., 2013 [15]	Sicilia	174	150
Vitalini et al., 2013 [16]	Lombardia	66	100
Di Sanzo et al., 2013 [17]	Basilicata	78	120
Tuttolomondo et al., 2014 [18]	Sicilia	108	230
Cornara et al., 2014 [19]	Liguria	120	52
Sansanelli et al., 2014 [20]	Emilia-Romagna	66	39
Dei Cas et al., 2015 [21]	Lombardia	126	92
Bellia et al., 2015 [22]	Piemonte	90	47
Fortini et al., 2016 [23]	Lazio	106	71
Vitalini et al., 2015 [24]	Lombardia	212	328
Sansanelli et al., 2017 [25]	Basilicata	52	58
Mautone et al., 2019 [26]	Campania	119	70
Lucchetti et al., 2019 [27]	Marche	92	120
Prigioniero et al., 2020 [28]	Southern Italy	524	831
Bottoni et al., 2020 [29]	Lombardia	59	137
La Rosa et al., 2021 [30]	Aegadian Islands	122	48
Danna et al., 2022 [31]	Aosta valley	217	68

**Table 2 plants-11-02246-t002:** Distribution of the 200 subjects interviewed in Trentino-South Tyrol based on age, sex, and occupation.

	N° Informants (%)				
Age	Male	Female	Student	Workers	Retired
<30	8 (12%)	20 (15%)	1 (1%)	27 (12%)	0
30–39	12 (18%)	17 (13%)	0	100 (50%)	0
40–49	6 (9%)	24 (18%)			
50–59	11 (16%)	30 (23%)			
>60	30 (45%)	42 (32%)	0	30 (15%)	42 (32%)

**Table 3 plants-11-02246-t003:** Distribution of the 200 subjects interviewed in Trentino-South Tyrol based on the 11 districts in which the study area is divided.

	N° Informants	N° Citations
Valle di Sole	12	62
Valle di Non	19	175
Valli Giudicarie	60	116
Alto Garda e Ledro	11	27
Valle dell’Adige	23	42
Vallagarina	15	56
Alta Valsugana	11	24
Bassa Valsugana e Tesino	14	67
Valle di Fiemme	15	148
Primiero	8	37
Ladino di Fassa	12	63
Tot	200	817

**Table 4 plants-11-02246-t004:** Number of citations for plants cited in the medicinal field divided into the category of use and related frequency relative citation (RFCs).

	Respiratory System	Integumentary System	Gastrointestinal System	Urogenital System	Musculoskeletal System	Nervous System	Mineralizing	Others	Total	RFCs
***Abies alba* Mill. **	3				3				6	**0.03**
***Achillea millefolium* L. **		2		18					20	**0.1**
** *Arctium lappa* ** ** *L* ** **.**		1	3						4	**0.02**
** *Arnica montana L.* **					94				94	**0.47**
***Artemisia absinthium* L. **			5						5	**0.025**
***Bryonia alba* L. **					2				2	**0.01**
***Calendula officinalis* L. **		21							21	**0.055**
***Capsella bursa-pastoris* L. **				2	1		1	1 ^a^	5	**0.025**
***Carum carvi* L. **			19						19	**0.095**
***Cetraria islandica* L. **	1								1	**0.005**
***Chelidonium majus* L. **		13							13	**0.065**
***Cornus mas* L. **			1						1	**0.005**
***Crataegus monogyna* Jacq. **								3 ^b^	3	**0.015**
** *Cyclamen sp.pl.* **								1 ^c^	1	**0.005**
***Elymus repens* (L.) Gould **				1					1	**0.005**
***Equisetum arvense* L. **				5			3		8	**0.04**
***Filipendula ulmaria* (L.) Maxim. **	1				16				17	**0.085**
***Frangula alnus* Mill. **			6						6	**0.03**
***Gentiana lutea* L. **			7				1		8	**0.04**
** *Hylotelephium telephium (L.) Holub* **		14							14	**0.07**
***Hypericum perforatum* L. **	3	69				3			75	**0.35**
** *Juniperus sp.pl.* **				1					1	**0.005**
** *Larix sp.pl.* **		16							16	**0.08**
***Leontopodium alpinum* Cass. **		4							4	**0.02**
***Lycopodium clavatum* L. **		1							1	**0.005**
***Malva sylvestris* L. **	11		14	5	2			2 ^d^	34	**0.17**
***Matricaria chamomilla* L. **		4	7	2	1			1 ^d^	15	**0.075**
***Melissa officinalis* L. **				9		3			12	**0.06**
***Nepeta cataria* L. **				1		1			2	**0.01**
***Oxalis acetosella* L. **			17					1 ^e^	18	**0.09**
***Papaver rhoeas* L. **	4					3			7	**0.035**
***Picea abies* (L.) H.Karst **	2								2	**0.01**
***Pinus cembra* L. **	16								16	**0.08**
***Pinus mugo* Turra **	69								69	**0.345**
***Plantago lanceolata* L. **	2								2	**0.01**
***Portulaca oleracea* L. **			1	1			3		5	**0.025**
***Potentilla reptans* L. **			5						5	**0.025**
***Pulmonaria officinalis* L. **	5								5	**0.025**
***Rhododendron ferrugineum* L. **					11				11	**0.055**
***Rodiola rosea* L. **						1	1		2	**0.01**
** *Rosa canina L.* **	5						4		9	**0.045**
***Sambucus nigra* L. **	10								10	**0.05**
***Satureja montana* L. **			26				1		27	**0.135**
***Senna alexandrina* Mill. **			1						1	**0.005**
***Solanum nigrum* L. **		1							1	**0.005**
***Solidago virgaurea* L. **				16					16	**0.08**
***Symphytum officinale* L. **					15				15	**0.075**
***Taraxacum* F.H.Wigg sect. *Taraxacum* **			6	7					13	**0.065**
** *Thymus sp.pl.* **	19								19	**0.095**
** *Tilia sp.pl.* **	16					2		1^b^	19	**0.095**
***Tussilago farfara* L. **	4								4	**0.02**
***Urtica dioica* L. **		10	4				8		22	**0.11**
***Vaccinium vitis-idaea* L. **				3					3	**0.015**
***Verbascum thapsus* L. **	1								1	**0.005**
***Viburnum lantana* L. **								2 ^d^	2	**0.01**
***Viola odorata* L. **	2								2	**0.01**
**Total**	174	156	122	71	145	10	22	12	712	

^a^ = Endocrine system.^b^ Cardiovascular system. ^c^ = Hearing system ^d^ = Visual apparatus. ^e^ = Circulatory system.

**Table 5 plants-11-02246-t005:** Plants mainly used for traditional preparations and related RFC values and medicinal applications.

#	Plants	RFC	Traditional Preparations	Medicinal Uses
**1**	** *Achillea millefolium* **	**0.1**	**Herbal tea**	**Menstrual pain and oligomenorrhea**
			**Decoction**	**Urinary tract infections**
			**Oleolyte**	**Muscle and joint pain**
2	*Arctium lappa*	0.02	Decoction	Diuretic
			Tincture	Muscle and joint pain
**3**	** *Arnica montana* **	**0.47**	**Oleolyte**	**Muscle and joint pain**
			**Oinment**	**Muscle sprains or contractures**
4	*Artemisia absinthium*	0.025	Herbal tea	Dyspepsia
5	*Bryonia alba*	0.01	Tincture	Muscle and joint pain
6	*Calendula officinalis*	0.055	Oleolyte	Skin diseases, wounds, and burns
7	*Capsella bursa-pastoris*	0.025	Tincture	Rheumatism
8	*Carum carvi*	0.095	Herbal tea	Dyspepsia
9	*Crataegus monogyna*	0.015	Herbal tea	Hypertension
10	*Cyclamen* sp.pl.	0.005	Oleolyte	Ear infection
11	*Equisetum arvense*	0.04	Herbal tea	Urogenital diseases
12	*Filipendula ulmaria*	0.085	Tincture	Muscle and joint pain
13	*Frangula alnus*	0.03	Herbal tea	Constipation
14	*Gentiana lutea*	0.04	Tincture	Dyspepsia
**15**	** *Hypericum perforatum* **	**0.35**	**Oleolyte**	**Skin diseases, wounds, and burns**
			**Oinment**	**Burns, erythema, or dermatitis**
			**Herbal tea**	**Respiratory infections and depression**
16	*Juniperus* sp.pl.	0.005	Herbal tea	Dyspepsia and urinary tract infections
17	*Larix decidua*	0.08	Oinment	Wounds healing
18	*Lycopodium clavatum*	0.005	Decoction	Muscle aches
**19**	** *Malva sylvestris* **	**0.17**	**Herbal tea**	**Dyspepsia**
20	*Matricaria chamomilla*	0.075	Herbal tea	Dyspepsia
21	*Melissa officinalis*	0.06	Herbal tea	Insomnia and premenstrual syndrome
22	*Nepeta cataria*	0.01	Herbal tea	Insomnia and premenstrual syndrome
23	*Papaver rhoeas*	0.035	Herbal tea	Insomnia and cough
**24**	** *Pinus mugo* **	**0.345**	**Mugolio**	**Diseases of the respiratory system**
25	*Plantago lanceolata*	0.01	Herbal tea	Catarrh
26	*Portulaca oleracea*	0.025	Herbal tea	Worm infection
27	*Potentilla reptans*	0.025	Herbal tea	Fever and diarrhea
28	*Pulmonaria officinalis*	0.025	Herbal tea	Cough
29	*Rhodiola rosea*	0.01	Herbal tea	Infections of the respiratory system
30	*Rhododendron ferrugineum*	0.055	Oleolyte	gout and joint pain
31	*Rosa canina*	0.045	Herbal tea	Cough and sore throat
32	*Sambucus nigra*	0.05	Herbal tea	Cold
**33**	** *Satureja montana* **	**0.135**	**Herbal tea**	**Nervous gastric pains, bloating, and vomiting**
34	*Senna alexandrina*	0.005	Herbal tea	constipation
35	*Solidago virgaurea*	0.08	Herbal tea	Urogenital diseases
36	*Symphytum officinale*	0.075	Oleolyte	Muscle and joint pain
37	*Taraxacum sect. Taraxacum*	0.065	Decoction	Diuretic
38	*Thymus* sp.pl.	0.095	Herbal tea	Infections of the respiratory system
39	*Tilia* sp.pl.	0.095	Herbal tea	Cough, cold, and tachycardia
40	*Tussilago farfara*	0.02	Herbal tea	Infections of the respiratory system
41	*Vaccinium vitis-idaea*	0.015	Herbal tea	Urogenital diseases
42	*Verbascum thapsus*	0.005	Herbal tea	Infections of the respiratory system
43	*Viola odorata*	0.01	Herbal tea	Cough

## Data Availability

Not applicable.

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
