# Peer review of "Walking around the Autonomous Province of Trento (Italy): An Ethnobotanical Investigation"

_plants, 2022, doi:10.3390/plants11172246_

Round 1
Reviewer 1 Report
The paper “Walking around the Autonomous Province of Trento (Italy): an ethnobotanical investigation” contributes to the growth of literature on the knowledge about the use of plants in natural medicine, and the ethnobotanical knowledge of the plants.
Before the manuscript acceptation for publication in “Plants” the following items should be revised:
Introduction
The description of the aim of the research is not clear.
Results
Table and Figure - the titles should be more specific
Figure 2 convert, for example, 87,10 to 87.10
In the statement of results, there is no scientific approach to these results - whether statistical analysis was used, e.g., the effect of age, sex or type of plant.
Line 200-201 “The so obtained preparation was conserved in glass jars (Figure 3 200 panel B, E).” Figure 3 includes panels A-D.
Figure 3. There is no explanation of the terms (A-D) in the title.
Figure 4 – 8. There is no literature for this information.
Methods
What was the method of selecting the group?
Conclusion
What are the weaknesses of this method?
Author Response
Introduction
The description of the aim of the research is not clear.
Authors: We thank the reviewer for his/her kind comments and suggestions. We rephrased the paragraph and added the following sentence at the end of the introduction to clarify our aim:
“Particularly, we selected and interviewed 200 subjects of different sex, ages and occupation to collect and analyze ethnobotanical information in the medicinal, veterinary and liquoristic fields.”
Results
Table and Figure - the titles should be more specific
Authors: All the titles have been changed and made more specific. Please find here below the new titles:
Table 2. Distribution of the 200 subjects interviewed in Trentino - South Tyrol based on age, sex, and occupation.
Table 3. Distribution of the 200 subjects interviewed in Trentino – South Tyrol based on the 11 districts in which the study area is divided.
Figure 2. (A) Percentage of citation collected during the interviews grouped for categories of use, and (B) main applications of the medicinal plants.
Table 4. Number of citations for plants cited in the medicinal field devided in category of use and related frequency relative citation (RFCs).
Figure 3. Common “in-house” packaging of traditional preparations: oleolyte (A and B), ointment traditionally called "ünt del tai" (C), and Mugolio (D).
Figure 4-8: literature citations have been added
Figure 2 convert, for example, 87,10 to 87.10
Authors: we thank the reviewer; Figure 2 has been corrected.
In the statement of results, there is no scientific approach to these results - whether statistical analysis was used, e.g., the effect of age, sex or type of plant.
Authors: We apologize for being not clear, and we are sorry for hearing that there is no scientific approach to the results. The most interesting plants have been selected basing on the RFC values and the traditional preparations emerged during the interviews. Moreover, the information collected derive from oral popular culture diffused horizontally in the population, so differences among age and sex was considered not relevant. As stated in the text, the only difference emerged is associated to the number of informants (female and old people are more numerous) and the reasons are explained in the text (progressive departure from traditional knowledge of male and younger people). Accordingly, statistical analysis is not applicable.
Line 200-201 “The so obtained preparation was conserved in glass jars (Figure 3 200 panel B, E).” Figure 3 includes panels A-D.
Authors: we thank the Reviewer for outlying the mistake. The error was due to a previous draft of the manuscript changed before the submission and it has now been corrected
Figure 3. There is no explanation of the terms (A-D) in the title.
Authors: the title of Figure 3 has been clarified and all the panels explained
Figure 4 – 8. There is no literature for this information.
Authors: we apologize for missing citations. Proper references have been added.
Figure 4 [45-47], Figure 5 [48, 49], Figure 6 [52-54], Figure 7 [60] and Figure 8 [61, 62]
Methods
What was the method of selecting the group?
Authors: Again, sorry for being not clear. At the beginning of the work the “perfect informant” was a figure e able, through an interview, to provide the information necessary to conduct the ethnobotanical investigation. It should be a subject over the age of 60, lived in the Autonomous Province of Trento and without particular qualifications of botanical studies. As the investigation proceeded, the limitation of this profile emerged. Consistently, other informants have been interviewed to enlarge the emerging information and the description of the selecting criteria was only described in the text as “Informants have been selected based on knowledge of traditional medicinal plants, and age.” (line 104).
Conclusion
What are the weaknesses of this method?
Authors: We thank the reviewer for his/her kind observation. In order to highlight the limitation of the proposed approach we added the following sentence at the end of the conclusion. “However, to confirm their therapeutic uses, a deeper investigation is needed to approve the safety and efficacy of the extracts and of their bioactive compounds.”

Reviewer 2 Report
This is a strong study and sounds good, but it is not valid for the Special Issue "Pharmacological and Toxicological Study of Medicinal Plants" of the journal Plants. This is just an ethnobotanical/medicinal study, not pharmacological or toxicological study. I am sorry. I suggest to submit it to more suitable journal, for example to Journal of Ethnopharmacology, or to Ethnobotany Research and Applications.
Author Response
Reviewer 2
This is a strong study and sounds good, but it is not valid for the Special Issue "Pharmacological and Toxicological Study of Medicinal Plants" of the journal Plants. This is just an ethnobotanical/medicinal study, not pharmacological or toxicological study. I am sorry. I suggest to submit it to more suitable journal, for example to Journal of Ethnopharmacology, or to Ethnobotany Research and Applications.
Authors: we thank the reviewer for finding appreciating our study. Regarding the journal, and the Special Issue in particular, we considered Plants Journal appropriate, considering that “Plants (ISSN 2223-7747) is an international and multidisciplinary scientific […] journal that covers all key areas of plant science. […] The main aim of our journal is to encourage scientists and research groups to publish theoretical and experimental results of research in all fundamental and applied fields of plant science.” Moreover, the keywords of the Special Issue “Keywords: medicinal plants; plants extract; phytotherapy; efficacy; toxicity; pharmacology; immunology; phytochemicals; bioactive compounds” are in line with the manuscript content.

Reviewer 3 Report
The manuscript entitled “Walking around the Autonomous Province of Trento (Italy): an ethnobotanical investigation” presents the survey on the medicinal plants used for human or animal health or for domestic purposes by inhabitants from Trentino - South Tyrol region.
The manuscript is very interesting and well-written. However, minor revisions should be made in order to be published in Plants journal, and the manuscript should be completed and/or modified taking into account the suggestions from attached file.

Author Response
Reviewer 3
The manuscript entitled “Walking around the Autonomous Province of Trento (Italy): an ethnobotanical investigation” presents the survey on the medicinal plants used for human or animal health or for domestic purposes by inhabitants from Trentino - South Tyrol region.
The manuscript is very interesting and well-written. However, minor revisions should be made in order to be published in Plants journal, and the manuscript should be completed and/or modified taking into account the suggestions from attached file.
The manuscript entitled “Walking around the Autonomous Province of Trento (Italy): an ethnobotanical investigation” presents the survey on the medicinal plants used for human or animal health or for domestic purposes by inhabitants from Trentino - South Tyrol region. The manuscript is very interesting and well-written. However, minor revisions should be made in order to be published in Plants journal, and the manuscript should be completed and/or modified taking into account the suggestions below:
- The authors are advised to rephrase the sentences from lines 20-21, 59-61, 143-144, 256- 259.
Authors: The sentence “Moreover, for each medicinal plant relative frequency citation, useful to identify the local importance of each species, was calculated and the main traditional preparations were discussed.” Was replaced as follows:
As a second step, for each medicinal plant local importance was evaluated by calculating their relative frequency citation value. Moreover, their main traditional preparations were discussed.”
The sentence “The present work aims to shed light on the ethnobotanical knowledge of Trentino Alto Adige, being the only written findings fragmentary, widely dispersed, and Italian written [32-34].” Was rephrased as follows:
“The present work aims to shed a light on the ethnobotanical knowledge of Trentino Alto Adige. So far, the available information is fragmentary, widely dispersed, and often guarded in oral popular culture or Italian written [32-34].”
The sentence “Informants reported different therapeutic applications of plants, depending on their route of administration. were different” was replaced with the following:
“Informants reported different therapeutic applications of plants, depending on their route of administration.”
The sentence: “Lipophilic metabolites may explain the use of oleolyte preparation (Table 5, #1) for the treatment of muscle and joint pain, (Figure 4) [46, 47], as well as apigenin, luteolin dicaffeoylquinic acids may explain the use of A. millefolium decoction for treating urinary tract infections (Table 5, #1).” was replaced with the following:
“The different hydro-lipophilic nature of the metabolites that are present in the phytocomplex extracted by A. millefolium may explain the different traditional preparations. Particularly, oleolyte (Table 5, #1) is able to extract lipophilic metabolites useful for treating of muscle and joint pain, (Figure 4), while decotion is exploited to extract apigenin, luteolin dicaffeoylquinic acids for treating urinary tract infections (Table 5, #1) [46, 47].
- The authors are advised to change „traditional preparations” (line 22).
Authors: the sentence has been changed
- The authors should check and correct the cited article [40] (line 114) which reffers to lichen Cladonia foliacea, and not Cetraria islandica.
Authors: the reference has been changed, as suggested
- I suggest the alphabetical order of plant species in table 4.
- The authors are advised to use Italic style for all plant species: Hylotelephium telephium - Table 4.
Authors: We formatted and corrected table 4 according to reviewer suggestions.
- The authors are advised to change „dried plant” (line 150), since usually not the entire plant is used, only one part, the vegetal product
Authors: We changed “dried plant” in “dried vegetal part”.
- The authors are advised to write which is the plant material used (line 181).
Authors: We added the aerial part in the text (galls or cecidia).
- The authors are advised to add details about each figure from Figure 3.
Authors: The titles have been changed and make more specific.
- The authors are advised to add the concentrations of traditional preparations used (if available) – table 5.
Authors: the concentration of actives in traditional preparation has not been specified by the informant. Traditional preparations have not been analyzed in this work. An analytical work is out of the scope of this specific manuscript. Nevertheless, we will take into account this suggestion of the Reviewer for a subsequent manuscript.
- In subsection 2.5, the presentation of plant species should be more specific, with several details about the chemical composition and traditional uses with focus on the scientific demonstration of traditional uses. Consequently, the reference list should be expanded.
Authors: In this manuscript we outlined the main characteristics and the main metabolites present in the most cited plant, with the aim to rationalize their traditional use. We inserted in the text (line 284) the following sentence "Along the years several reviews regarding these plants have been published, and they can be useful to readers interested in deepening the knowledge about the above discussed medicinal plants.” We thank the reviewer for the suggestion: we will write a comprehensive review on such plants, specifically focused on their use in Tentino Alto Adige region.
- Line 293: Hypericum - is a genus with several species, about which species the authors write?
- The authors are advised to correct the term “mucilaginous” (line 334)
Authors: The reviewer is right; we correct the errors.
